Analysis of brain activation and wave frequencies during a sentence completion task: a paradigm used with EEG in aphasic participants

Lima Claudia 1 claudia.lima@fieb.org.br
Lopes Jeferson Andris 2
Souza Victor 3
Barros Sarah 4
Winkler Ingrid 5
Senna Valter 1
1 MCTI, Senai Cimatec University Center , Salvador, Bahia , Brazil
2 Brasilia Technical School , Brasília, Federal District , Brazil
3 São Rafael Hospital , Salvador, Bahia , Brazil
4 Neurological Rehabilitation Sector, CEPRED , Salvador, Bahia , Brazil
5 GETEC, Senai Cimatec University Center , Salvador, Bahia , Brazil
Patton Bob
Electronic publication date: 2023 Jun 12
Publication date: 2023
Volume: 11
Electronic Location ID: e15518
Received 2022 Dec 14; Accepted 2023 May 15
Copyright: © 2023 Lima et al.
Copyright year: 2023
Copyright holder: Lima et al.
License: This is an open access article distributed under the terms of the Creative Commons Attribution License, which permits unrestricted use, distribution, reproduction and adaptation in any medium and for any purpose provided that it is properly attributed. For attribution, the original author(s), title, publication source (PeerJ) and either DOI or URL of the article must be cited.
License URL: https://creativecommons.org/licenses/by/4.0/

Keywords: Brain activation, Aphasia, Electroencephalography, Language rehabilitation

Funding: Bahia State Research Support Foundation (FAPESB) process 0195/2019 National Council for Scientific and Technological Development (CNPq) Proc. 308783/2020-4 This work was suported by the Bahia State Research Support Foundation (FAPESB), process 0195/2019 and the National Council for Scientific and Technological Development (CNPq), Proc. 308783/2020-4. The funders had no role in study design, data collection and analysis, decision to publish, or preparation of the manuscript.

==============================
Aphasia is a language disorder that occurs after brain injury and directly affects an individual’s communication. The incidence of stroke increases with age, and one-third of people who have had a stroke develop aphasia. The severity of aphasia changes over time and some aspects of language may improve, while others remain compromised. Battery task training strategies are used in the rehabilitation of aphasics. The idea of this research is to use electroencephalography (EEG) as a non-invasive method, of electrophysiological monitoring, with a group of aphasic patients in rehabilitation process in a prevention and rehabilitation unit of the person with disabilities of the Unified Health System (SUS), of reference in the state of Bahia-Brazil. In this study, the goal is to analyze brain activation and wave frequencies of aphasic individuals during a sentence completion task, to possibly assist health professionals with the analysis of the aphasic subject’s rehabilitation and task redefinition. We adopted the functional magnetic resonance imaging (fMRI) paradigm, proposed by the American Society for Functional Neuroradiology as a reference paradigm. We applied the paradigm in the group of aphasics with preserved comprehension, right hemiparesis, and left hemisphere injured or affected by stroke. We analyzed four electrodes (F3/F4 and F7/F8) corresponding to the left/right frontal cortex. Preliminary results of this study indicate a more robust activation in the right hemisphere (average of aphasics), with a difference of approximately 14% higher in Theta and Alpha frequencies, with 8% higher in low Beta (BetaL) and with approximately 1% higher in high Beta frequency (BetaH), Gamma frequency was higher by approximately 3% in the left hemisphere of the brain. The difference in electrical activation may be revealing to us a migration of language to the non-language dominant hemisphere. We point to possible evidence suggesting that EEG may be a promising tool for monitoring the rehabilitation of the aphasic subject.

Introduction

Due to a local brain injury, a person may not be able to speak and understand the people around him, or he may understand but not be able to make himself understood. In other cases, they can make sentences, but do not use articles and connecting terms, which leaves the message confusing. Sometimes, they lose the ability to name things, objects, and have no command of the word, for example to name an object, such as table or an animal, as horse. This means that the brain injury has affected the area responsible for language control, and the symptoms vary from person to person.

This is aphasia, a language disorder that affects a person’s ability to communicate, limits social interaction, contributes to isolation, depression, and heavily impacts quality of life.

The most common cause of aphasia is stroke and traumatic brain injury (TBI), mainly in the left hemisphere, where language function is usually located. Approximately one-third of stroke people develop aphasia (Brady et al., 2016; Mattioli, 2019).

“The incidence of stroke increases with age and is higher in men than in women” (Mattioli, 2019) and the severity of aphasia “can change over time and single aspects of language impairment may improve while others remain compromised”. Aphasia dramatically affects the participant’s and their families functional communication, daily activities, and social skills. Therefore, effective rehabilitation is essential.

A study evaluated the effects of speech and language therapy (SLT) for aphasia after stroke (Brady et al., 2016), and concluded that “therapy at high intensity, high dose, or over a longer period may be beneficial”. However, it warns that high-intensity and high-dose interventions may not be acceptable for all participants.

There is a wide variability of language tests that can be applied in the rehabilitation process of aphasics; however, current evidence suggests that monitoring the aphasic is very difficult to achieve. This leaves us with an open question of which method we should use.

In the study by Sreedharan et al. (2020), real-time functional magnetic resonance imaging (RT-fMRI) was used as a neurofeedback training strategy, to improve neural activation in the language areas of post-stroke participants, with expressive aphasia, to improve language deficits. For the author, this strategy is very expensive since there are many hours for training, and the system is not portable, making the cost very high. However, he claims that it is possible to adjust neurofeedback training to activate specific study regions using less expensive and portable methods, such as EEG-based systems, which may be feasible for rehabilitation.

Another essential strategy for studying the brain, according to Black et al. (2017) is the “sharing, comparison and generalization of results”. According to the authors, institutions perform fMRI in markedly different ways. With this challenge, a task force from the American Society of Functional Neuroradiology members from various institutions created “two sets of standard language paradigms that strike a balance between ease of application and clinical utility”. An adult language paradigm for pre-surgical language assessment includes the sentence completion task and another paradigm geared toward children (Black et al., 2017).

Given this scenario, the present study proposes to apply the fMRI paradigm, developed in the study of Black et al. (2017) with EEG, to analyze the brain activation of aphasic individuals with the task of completing sentences or phrases. This study was developed at the Center for Prevention and Rehabilitation of People with Disability (CEPRED) which is a unit of the Brazilian Unified Health System (SUS) of reference in the state of Bahia.

With the perspective of application in other rehabilitation and clinical units, this research has three pillars: daily accessibility (cost and mobility of the equipment), usability (ease of use of the therapy equipment), and whether the technology is invasive or not for the participant.

The monitoring of rehabilitation, throughout the therapeutic process, generally occurs with the repetition of language tests and the evolution of responses to these tests. There is a lack of studies that analyze the applicability of accessible instruments, outside the hospital environment, that can support the health professional in monitoring aphasia rehabilitation.

We hypothesize that by investigating the electrical activation of the brain and the wave frequencies during the execution of language tests, we can analyze whether there is activation in a non-dominant area of language. The method can support improving the therapeutic process by monitoring the rehabilitation of the aphasic.

Literature review

Aphasia

Aphasia is a language disorder caused by an encephalic injury or dysfunction, accompanied or not by cognitive changes. It is considered “one of the most common neurological alterations following focal lesion acquired in the central nervous system, in areas responsible for comprehensive and/or expressive language, oral and/or written” (Kunst et al., 2013).

For Brady et al. (2016), aphasia is “a language impairment acquired after brain damage affecting some or all language modalities: speech expression and comprehension, reading, and writing”. Other diseases can also cause aphasia, such as tumors, trauma, degenerative or metabolic diseases.

Aphasia has been “a historical target of investigation and scientific debate in medicine, neuropsychology, and linguistics” (Mineiro et al., 2008). There are eight types of aphasia: Broca’s aphasia (or expressive aphasia), Wernicke’s aphasia, conduction aphasia, global aphasia, transcordial motor aphasia, transcordial sensory aphasia, mixed transcordial aphasia, and anomic aphasia.

For Mineiro et al. (2008), the classification of aphasias is based on the participant’s “performance in certain parameters that are evaluated through batteries of tests”, and the most used parameters are: speech fluency, ability to understand orders, ability to name objects, and ability to repeat words.

For the prognosis of aphasia, Plowman, Hentz & Ellis (2012) emphasize that this is a daunting task and involves considering the location and size of the injury, the type of aphasia, the person’s age and gender, and other elements focused on family support, motivation, and medical care.

Regarding lesion location associated with aphasia type, Bocquelet et al. (2016) states that “lesions of ventral temporal lobe flow regions result in Wernicke’s aphasia characterized by impaired speech comprehension, while lesions of frontal areas result in Broca’s aphasia characterized by impaired speech production”.

Diagnoses of aphasia are obtained through neurological evaluation and brain imaging tests, such as magnetic resonance imaging (MRI) or computed tomography (CT) (Pommerehn, Delboni & Fedosse, 2016). The assessment process involves linguistic examination instruments and should “address different levels and linguistic components, including comprehension and expression and the use of language at the levels of word, sentence, and speech” (Pagliarin et al., 2013).

Also, according to Pagliarin et al. (2013), in a study conducted to evaluate the panorama of national and international literature about language examination instruments used after left hemisphere brain injury (LHE), they conclude that “efforts are needed for the development and standardization of language instruments for the assessment of adult participants with LHE and other neurological conditions suitable to the Brazilian sociocultural reality”. They also draw attention to the scarcity of studies with such instruments specifically involving participants with LHE.

The study developed by Casarin et al. (2011) also revealed a scarcity of instruments (or assessment tools of functional language components) in the national context and concluded that there is a need for new studies to fill this gap. They add that “even at the international level, there seems to be an unmet demand for clinical tools for a brief assessment of functional language components, such as pragmatic-inferential and discourse”.

Hill, Kovacs & Shin (2015) points to the challenge of promoting the interconnection of a brain-computer interface (BCI) with augmentative and alternative communication (AAC), using brain signals as an alternative for a “non-invasive approach to access assistive technologies”. According to the same authors, conventional alternative communication is satisfactory when it requires a certain degree of voluntary muscle control, although AAC is “vital for helping people with complex communication”.

Brain-controlled technologies appear to be even more promising in aiding communication for aphasic individuals, minimizing the consequences of the physical limitations that many of them have. Brain-computer interfaces (BCI) can aid communication in disabled populations by utilizing neurobiological signals (Brumberg et al., 2010).

Strategies have been developed for BCI’s such as “letter selection” systems, which establish indirect communication with hand, arm, or eye movements, which can be decoded. For Bocquelet et al. (2016), “the creation of a speech BCI to restore continuous speech directly from the neural activity of speech-selective brain areas are an emerging field in which increasing efforts need to be invested”.

Brain-computer interfaces seem promising for “enabling artificial speech synthesis from continuous decoding of neural signals underlying speech imagery” (Bocquelet et al., 2016). These interfaces, according to the same authors, do not exist yet, and for their design it is essential to consider three points: “the choice of appropriate brain regions to record neural activity, the choice of an appropriate recording technique, and the choice of a neural decoding scheme in association with an appropriate speech synthesis method”.

Testing, techniques, and language paradigm

Speech therapy treats language disorders and their therapy. Speech and language therapy (SLT) is a “complex rehabilitation intervention aimed at improving language and communication skills (verbal comprehension, spoken language, reading and writing), activities, and participation” (Brady et al., 2016).

A study conducted by Altmann, da Silveira & Pagliarin (2019) on “speech therapy intervention in expressive aphasia” reveals that “among the traditional therapies found, the following were observed: word retrieval therapy, melodic therapy, and conversational therapy”. The authors conclude that “word retrieval therapy was the most used traditional method” and “figure naming and figure/word relationship were the most used strategies,” the authors emphasize that “the majority used figure naming, and the use of tablets and computers was frequent, mainly as a means of presenting the stimuli”.

Research developed by Pagliarin et al. (2013) sought to identify “language investigation instruments used for the evaluation of sudden onset neurological pictures involving the left hemisphere (LHE)” and analyze which linguistic components are the most evaluated. As a result of this study, the authors identified “nine internationally used instruments that assess different language components in participants with LHE”, highlighting the Boston Diagnostic Aphasia Examination (BDAE) as one of the “most internationally used instruments for aphasia detection”. The most investigated linguistic components were “naming and listening comprehension”. Finally, the authors reveal that the “instruments used internationally present appropriate adaptations and standardizations for the social, cultural and linguistic reality of the evaluated population”.

In general, for speech and language evaluation, to confirm a diagnosis of aphasia, the phonoaudiologist analyzes the following areas: naming, repetition, language comprehension, reading, and writing. In addition to speech and language evaluation, diagnostic imaging tests can be performed using an MRI or CT scan.

Black et al. (2017) proposed the adoption of a standard language paradigm, to be used with fMRI as the first step to advance knowledge sharing. The authors hope that by adopting the reference paradigm, with a battery of language tasks, among them the “complete sentences” task, validated by a committee of experts, it will be possible to improve participant care.

Also, according to Black et al. (2017) the Sentence Completion Task (T1) produces greater activation in both temporal and frontal regions, “suggesting a more robust activation of language networks because it combines language comprehension as well as production in a naturalistic way”.

For Zapała et al. (2018) the main challenges lie in the uneven learning pace of each individual and the effective operation of BCI devices, which use many techniques, e.g. fMRI, fNIRS, EcoG, MEG and EEG, for recording brain activity. Some of these techniques (or methods) are very expensive to implement or are very robust, making mobility difficult.

However, electroencephalography (EEG) represents the basis of about 60% of the systems tested. Electrophysiological recording can offer adequate temporal resolution to track brain activity at the scale of speech production dynamics (Bocquelet et al., 2016).

Studies have revealed to us that, in addition to analyzing brain activity and neurological problems, it is possible to investigate neuronal patterns that may be associated with certain diseases. To improve the recognition of brain activity patterns in BCI applications, a recent study conducted by Zhang et al. (2021) proposed a clustering-based multitask learning algorithm to optimize the EEG feature. For the authors, “unlike single-task learning, multitask learning allows us to jointly select the most significant features of multiple tasks”.

Liang et al. (2022) study, meanwhile, concluded that “emerging matrix learning methods achieved promising performances in electroencephalogram (EEG) classification”. It points out that “the methods usually need to collect a large amount of individual EEG data and this, in general, causes fatigue and inconvenience to the participants”.

Chen et al. (2022) emphatically reveals to us that “in recent years, machine learning (ML) and artificial intelligence (AI) have become increasingly popular in the analysis of complex neural data patterns”. He also points out that EEG and fMRI are two rich neuroimaging modalities, and that these allow us to comprehensively investigate brain functions. He concluded that “advances in modern ML-powered technologies will create a paradigm shift in current practice in diagnosis, prognosis, monitoring, and treatment of mental illness”.

Given this scenario, it is understood that it is essential to seek a method that will support health professionals in the ongoing rehabilitation process of aphasic people, considering the applicability in rehabilitation centers.

Materials and Methods

The method proposed in this study, shown in Fig. 1, uses the standard paradigm developed by the American Society for Functional Neuroradiology (Black et al., 2017), with EEG, to analyze the brain activation of aphasic participants while executing sentence completion tasks.

Figure 1 Design of the experiment procedure.

Population and sample

The present research was approved by the Ethics Committee on Human Research of the Integrated Manufacturing and Technology Campus (CIMATEC)—Senai/Bahia (CAAE: 29622120.2.0000.9287) and approved by the Health Secretariat of the State of Bahia—SESAB (CAAE: 29622120.2.3001.0052) with the Center for Prevention and Rehabilitation of People with Disabilities-CEPRED, as a co participant center.

The study population was constituted of participants diagnosed with post-stroke aphasia, according to the report in medical records, who were in rehabilitation treatment and recruited by the speech therapy team of CEPRED, in the city of Salvador-Bahia, following the inclusion and exclusion criteria.

The sample was made up of eleven aphasic participants (Ap = Eight women, three men). The average age of all participants was 54 ± 7 years. Of these participants, 10 people with brain damage in the left hemisphere and one person with right hemisphere damage, all with hemiparesis (difficulty moving half of the body). A description of the volunteer participants is shown in Table 1. None of the participants had hearing impairment. They signed informed consent to participate in the study.

Table 1 Description of the participants.

Patient ID	Gender	Age	Hemiparesis	Hemisphere affected	Time of stroke (year/month)	Type of Aphasia	
Ap01	F	53	Right	Left	03/03	Broca (Expression)	
Ap02	F	51	Right	Left	06/11	Broca (Expression)	
Ap03	F	53	Right	Left	02/01	Anomic	
Ap04	F	45	Right	Left	01/05	Transcortical motor	
Ap05	F	63	Right	Left	04/05	Broca (Expression)	
Ap06	F	45	Right	Left	01/03	Transcortical motor	
Ap07	M	58	Right	Left	01/07	Broca (Expression)	
Ap08	M	63	Right	Left	00/05	Broca (Expression)	
Ap09	F	59	Right	Left	10/11	Broca (Expression)	
Ap10	F	45	Left	Right	05/03	Global	
Ap11	M	60	Right	Left	01/03	Global	

Inclusion criteria: participants of both genders, aged over 18 years, diagnosed with post-stroke aphasia and normal or normal-corrected vision.

Exclusion criteria: participants who had mental disorders, identified by a healthcare professional were not invited to the study. Participants with unstable cardiovascular disease or other serious diseases that impeded them from performing the tasks were also not part of the research.

The instrument: EEG

For this study, we used the Emotiv Epoc+, 16-channel, EEG Brainwear with Brain Computer Interface (BCI) technology. Designed for human brain research, from Emotiv based in San Francisco, USA, to capture EEG signals, sampled at 128/256 Hz. Commercially available and validated by other researchers (Badcock et al., 2013; Yu & Sim, 2016; Kotowski et al., 2018; Fouad, 2021; Melek, Melek & Kayikcioglu, 2020).

The device (headset type) is flexible and wireless, with the electrodes placed according to the international 10–20 positioning system, which is the standard international arrangement for electroencephalographic analysis.

Many studies have used the portable electroencephalography (EEG) system outside the hospital environment because it is less expensive, easy to set up, and comfortable. This device is opening doors to a wide range of studies, offering promising potential in rehabilitation.

One of the great advantages of EEG is the high temporal resolution, which reaches an accuracy of milliseconds, when compared with techniques such as fMRI. Figure 2 shows the location of each of the 16 sensors and the anatomical correspondence of the brain area. The odd-terminated channels are in the left hemisphere and the even-terminated channels are in the right hemisphere, which are: antero-frontal (AF3, AF4), frontal (F3, F4, F7, F8), front-central (FC5, FC6), temporal (T7, T8), parietal (P7, P8) and occipital (O1, O2). The mastoid sensor (M1) acts as a reference point to which the voltage of all other sensors is compared. The other mastoid (M2) is a feed-forward reference that reduces external electrical interference.

Figure 2 Location of electrodes on the 10–20 system and anatomical correspondence of brain areas (source: Epoc+ specifications).

Sentence completion task

We used the fMRI paradigm for adults, with adaptations for the Portuguese language. We will present in this study the results of the “Sentence Completion Task”, here called Task 1 or simply T1, with duration of 4 min (240 s), where each volunteer participant performed one experimental session.

About the T1 stimuli, the participant sees, on a video monitor, an incomplete sentence (example: Whales and dolphins live at …….) and is asked to mentally respond, just think the word (without speaking or moving lips). While the sentence is on the screen (5 s), the participant must continue to think of alternative words that complete the sentence until the next sentence appears. Then, there are three more phrases to be completed mentally.

After the end of the stimulus time (20 s), the “control” of the task begins, where the participant sees nonsense sentences (a set of letters separated by spaces) at the same time and quantity of repetitions. Thus, there are four stimulus sentences and four control sentences, interleaved in this order until the 4 min are completed. The task has a total of 24 stimulus sentences and 24 control sentences.

The researcher read sentence by sentence aloud to the participant who had little education. This procedure activates Wernicke’s area that is in the left hemisphere, specifically in the temporal lobe, which is the area responsible for understanding and receiving language, meaning that “we can activate it without necessarily speaking or communicating, simply by listening or reading” (Ardila, Bernal & Rosselli, 2016). Performing the sentence completion task increases the activation of both the temporal region and the frontal region, activating Wernicke’s area and Broca’s area (Black et al., 2017).

Details of the method

For the execution of the experiment, the participant was comfortably seated in front of a computer screen. The research protocol was read, informing that the participant is there voluntarily to participate in the study, which aims to investigate aphasia in expectation of supporting rehabilitation. Next, it was confirmed that the participant signed the free and informed consent form, which contained the details of the research, the researcher’s contact information, and the confidentiality of the participant’s personal data.

The method has five stages, detailed below:

The first stage: Pre-test. This was created to explain to the participant how the task would be executed. We presented a PowerPoint file with examples of the task, giving them the opportunity to test and ask questions before starting to collect the electroencephalographic signals.

The second stage: Data collection. To capture and record the electrophysiological signals, the EEG headset was placed on the participant’s head, with the antero-frontal electrodes (AF3 and AF4) positioned at a distance of three fingers above the participant’s eyebrow, allowing easier adjustment of the other electrodes in their respective areas.

With the EmotivPro 3.3.0.433 software, the connection of all electrodes was checked. When correctly positioned, they turned green on the program’s interface. If any electrode did not show the green color, some adjustments were necessary, such as removing hair strands under the electrode or performing new hydration with saline solution, as recommended by the manufacturer. The signal collection only started after the correct connection of all the electrodes. The participant was instructed not to move his head, arms, legs, and hands to reduce the occurrence of unwanted noise.

On the day of the collection, the participants were instructed to have their hair washed and dry, without using moisturizing creams, oils, or similar cosmetics. For participants with voluminous hair, it was requested that, if possible, their hair be braided to facilitate the placement of the headset and electrode contact.

Data collection begins with the execution of the baseline protocol (from the EEG device), which lasts 40 s in total. The protocol starts with a 5 s count and then an image appears indicating for the participant to keep his eyes open for 15 s. This is followed by a 5 s countdown and an image indicating that the participant should close his or her eyes. With 15 s countdown the participant should keep his eyes closed. When the baseline protocol is finished, the researcher presses key 1 to mark the beginning of the sentence completion task.

The acquisition of brain signals is finished when the participant completes the last sentence (4 min defined in the task). The researcher closes the session and exports the raw data in CSV (Comma-separated values) format.

We used four electrodes positioned on the frontal lobe, two electrodes on the left hemisphere (F7, F3) and two electrodes on the right hemisphere (F4, F8) and investigated five frequency ranges: Theta (4–8 Hertz), Alpha (8–12 Hertz), Low Beta (12–16 Hertz), High Beta (16–25 Hertz) and Gamma (25–45 Hertz).

Third stage: Importation of the electroencephalographic signals. For steps 3 and 4 we used MATLAB since it is a tool and a high-level programming language, widely used for signal processing by millions of engineers and scientists to analyze data, develop algorithms and create models. We used MATLAB version 9.12 (R2022a; MathWorks, Inc., Natick, MA, USA) where we worked with calculus, matrices, signal processing and graph construction. We developed scripts to import the electroencephalographic signals (CSV files) from all the participants.

After that, we checked the beginning of the task (with the registration of the numeral 1), so that we could separate the electrophysiological registers before the “start” of the task. Therefore, the raw data were loaded and stored in vectors.

Next, we converted the signal to microvolts (uV) and then take the DC level of the signal (offset). The asymmetric response to the fault is called DC Offset, which is a natural phenomenon of the electrical system. Afterward, we plot graphs of the channels in time and frequency. We follow with the generation of the spectrum at frequency, applying the algorithm that calculates the Discrete Fourier Transform (DFT) and its inverse (inverse Fourier Theorem). In summary, we applied the Fast Fourier Transform (FFT) so that the raw EEG signal could be identified as distinct waves with different frequencies. The detail of the data processing scheme is shown in Fig. 3 and includes pre-processing and post-processing.

Figure 3 Data processing schematic.

Fourth stage: Signal processing. We use the Signal Processing Toolbox™ and the Designfilt function for filter testing. We tested FIR (Finite Impulse Response) and IIR (Infinite Impulse Response) filters on the data, analyzing the best option for removing noise and artifacts (such as muscle, motion, sweating artifacts). We also tested the following filters: a) Low-pass filter: A low-pass filter allows low frequencies to pass through without difficulty and attenuates the amplitude of frequencies higher than the cutoff frequency.

b) Band-pass filter: A band-pass filter allows frequencies in a certain range to pass through and rejects frequencies outside this range.

c) High-pass filter: A high-pass, or high-pass filter, allows high frequencies to pass through easily, and attenuates the amplitude of frequencies below the cutoff frequency.

We subsequently created “windows” by fragmenting the signal every 5 s (Fig. 4). The window creation time was determined by the time a sentence is on the video monitor as a stimulus for the participant. We used the calculation of the RMS value of the signal in each window (the effective value, also known as the RMS value from the acronym Root Mean Square). For post-processing, we developed scripts called “Brain Activation” and “Band-Power”, detailed in the results section.

Figure 4 Examples of EEG recording windows (electrode F4) and the frequencies.

(A) Aphasic participant 2, Window 1. (B) Aphasic participant 8, Window 33 with artifact.

The fifth stage: Analysis of the results. The health professional, based on the results obtained, will adjust, or maintain the language stimulation conducted in the next sessions.

Results

From the 11 participants evaluated, regarding the determination of the areas of greater brain activation and wave frequencies by EEG, problems occurred in the performance of two participants (Ap10 and Ap11). These two participants went through step 1 of pre-test (Fig. 1) and verbalized that they understood the task, we moved on to step 2, for connection and data collection. After connecting the electrodes, we started the “Baseline Protocol”, with eyes open and then with eyes closed.

However, we noticed that both participants seemed to misunderstand the baseline protocol. During the performance of the task, these participants frequently shifted their focus of attention to look at their hands, to objects in the room, and verbalized a few words. These two participants had a diagnosis of global aphasia, with deficits in expression and in comprehension; due to this typology and the degree of distraction, it was not possible to include the data collected in our sample. The absence of comprehension is common in global aphasics. Therefore, for this study we considered only aphasics with preserved comprehension as shown in Table 1.

After importing the raw data, we did the pre and post processing of the data (Fig. 3). In the preprocessing, we did the filtering process, as explained in the materials and methods section, and used the strategy of creating “windows” that is used by health professionals for visual inspection and selection of windows without artifacts.

With the use of the windows, we were able to have the first view (view of each individual of the sample) of how the electrical activation occurred in each of the four electrodes (F7, F3, F4 and F8), verify the frequencies (Delta, Theta, Alpha and Beta), and disregard the windows with artifacts (represented in Fig. 4B with a red line).

In the post-processing step, we calculated the average value of each wave and normalized the power per electrode. This allowed us to have a comparative view of the frequencies, participant by participant, as shown in Fig. 5.

Figure 5 Examples of brain activation by electrode (normalized data).

(A) Aphasic participant 1. (B) Aphasic participant 2. (C) Aphasic participant 5. (D) Aphasic participant 7.

We hid the Delta wave because it is recorded more frequently in infants and children, and because it is related to involuntary body movements such as breathing, heartbeat, and digestion, therefore focusing our analysis on the language task.

In Fig. 6, we have examples of aphasic participants and their electrical activation (difference) between hemispheres. For ease of understanding, we considered the left hemisphere to be the negative values (bars down) and right hemisphere to be the positive values (bars up). The value shown on the y-axis represents the difference, in percentage, of the electrical activation between the two hemispheres of the brain.

Figure 6 Examples of the difference in electrical activation between hemispheres.

(A, B) Greater electrical activation in the right hemisphere, at all frequencies. (C) Greater electrical activation in the left hemisphere, at all frequencies. (D) Greater electrical activation in the right hemisphere at the Theta and Alpha frequencies and with greater electrical activation in the left hemisphere at BetaL, BetaH and Gamma.

Thus, we verify that there was a significant difference of electrical activation in the brain hemispheres. To exemplify, in Fig. 6B we observe that the frequency Theta, alpha, BetaH, and Gamma had greater electrical activation in the right hemisphere (bar up) with a difference between the hemispheres of 25% more. In addition, the BetaL frequency showed a difference of more than 30%.

In five participants we found higher electrical activation in the right hemisphere, at all frequencies (Theta, Alpha, BetaL, BetaH, Gamma), as examples in Figs. 6A and 6B. Two participants (AP08 and AP09) showed higher electrical activity at the frequencies Theta, Alpha and BetaL, also in the right hemisphere. One participant (Ap07) had greater activation at the Theta and Alpha frequencies in the right hemisphere (Fig. 6D), and a single participant (Ap05) had greater activation, in all frequencies, in the left hemisphere (Fig. 6C).

These data show us that eight in nine of the participants showed increased electrical activation in the non-language dominant hemisphere. These results may be reflecting language migration, counter-lateral processing.

Figure 7 shows the difference in electrical activation between brain hemispheres, considering the average of aphasic participants. Only the Gamma frequency had the highest activation in the left hemisphere, with approximately 4%. This wave frequency is associated with the processing of auditory, tactile, and visual stimuli. The other frequencies (Theta, Alpha, BetaL and BetaH) had greater activation in the right hemisphere.

Figure 7 Difference in electrical activation between the cerebral hemispheres (average of aphasic participant).

In two male participants (Ap07 and Ap08) we found less electrical activation in the right hemisphere, which may be showing us a lower migration of language to the right side, example in Fig. 6D.

Therefore, the “complete sentences” task was effective in activating language areas, using an EEG, in both hemispheres. These results converge with Lam et al. (2016) study, which used magnetoencephalography (MEG), with the sentence completion task, and emphasizes that the task “recruits areas distributed in both hemispheres and extends beyond classical language regions”.

Our study brings an important contribution to aphasia rehabilitation programs, regarding the use of EEG to identify the areas of greatest brain activation in task execution, during rehabilitation. These results may be highlighting that acquired lesions can cause language to reallocate in the opposite hemisphere.

Discussion

The objective of this study is to investigate the electrical activation of the brain and the wave frequencies during the execution of a sentence completion task, with the expectation of helping health professionals, especially speech and hearing therapists, in the rehabilitation of aphasic participants and the improvement of language deficits.

According to the method adopted, we did not include in our analysis the data from participants with global aphasia. We considered aphasic participants with preserved comprehension. Right-sided hemiparesis was observed in these participants and confirmed in hospital discharge records. From this, we inferred that the brain lesion and the dominant hemisphere of language were both located on the left side of the brain. However, based on the data collected and analyzed, we found increased electrical activity in the non-language dominant hemisphere during the execution of the task. This result may be revealing the migration of language to the contra-lateral hemisphere.

Our results are in line with the study by Morais (2020) who emphasizes that the brain may suffer “translocation of primary functions, especially when the lesion is located in the dominant hemisphere” even though the left hemisphere is considered the “higher language processor”, characterized as verbal, analytical, and intelligent (Harrington, 1989). Therefore, for this study, the translocation (movement or change of something from one location to another) of language may indeed have occurred to the right hemisphere, since the language of these aphasics is allocated to the left (dominant) hemisphere.

Regarding the two male participants (Ap07 and Ap08), we found less electrical activation in the right hemisphere, which we believe reveals less language migration and perhaps less recovery. There is a convergence here with other studies that suggest that there may be “a difference in language impairment and recovery rate after stroke, with women outperforming men” (Halpern, 2000).

Although our sample has a predominance of female participants, we cannot say that the result has this bias. We intend to continue with data collection, increasing the sample that will enable us to apply in-depth statistical analysis.

For Kimura (1983), when comparing the female and male sexes, there is a difference in speech organization and in the activities of the left hemisphere of the brain. Núñez et al. (2018) calls attention that the female brain, in general, is more symmetrical than the male brain and emphasizes that several structures have been identified as more symmetrical and are involved in language production.

Previous research has shown that sequential task batteries are used in the rehabilitation of aphasics but neglect the possibility of “rehabilitation traceability”, that is, of doing a follow-up and analysis of data collected during the rehabilitation process over time. This may be due to the lack of experimentation with accessible techniques in a rehabilitation environment.

The experimental procedure shown in our work adds this capability and indicates that an EEG device can adequately capture brain signals, providing a perspective for monitoring the rehabilitation of the aphasic. A device already recognized for providing adequate temporal resolution to track brain activity at the scale of speech production dynamics (Bocquelet et al., 2016).

We have defined a strategy that can be used by healthcare professionals in rehabilitation settings outside of hospitals. By using the experimental procedure in this study and seeing that a certain frequency is changing, the health-care professional will be able to monitor the rehabilitation, verifying if the participant is recovering better or not.

That said, we adopted the reference paradigm proposed by the American Society for Functional Neuroradiology as a standard language paradigm that strikes a balance between ease of application and clinical utility. It was developed as a strategy for “sharing, comparing, and generalizing results”. The literature shows us that institutions conduct their research with different methods and techniques, and this is a complicating factor in studying the brain (Black et al., 2017). These factors were essential for our study developed at CEPRED as we needed to structure a procedure that was user-friendly, portable, and non-invasive.

As advantages of our method, we highlight three aspects: accessibility, usability, and non-invasiveness. Regarding accessibility, our method has a lower cost and greater mobility since it is portable when compared to fMRI. Regarding usability, it is easy to use by health professionals, especially speech-language pathologists. The third advantage is the fact that the electrophysiological monitoring method is not invasive to the participant.

As a limitation, the proposed method was restricted to the analysis of aphasic participants with preserved comprehension. The other participants with disturbed comprehension were unable to follow the guidelines and therefore did not have their data included in the result.

The notable limitation in this study was our sample size, which was attributed to the complexity of the participants’ condition, most of whom had depression, sadness, and limitations in moving around. Therefore, our results may not be applicable to all aphasic participants and do not preclude the possibility of using other techniques.

The preliminary results obtained may guide the construction of new therapeutic strategies and may help in the monitoring of aphasic participants. As a near future perspective, we hope to increase our sample size, extend the analysis to the other lobes of the brain, and include three more tasks (word generation, rhyme, and object naming). We also hope that other studies can continue this line of investigation, working with the application of artificial neural networks and artificial intelligence.

Conclusions

We used EEG, which allows the recording of electrical currents emitted by the brain through electrodes applied on the scalp, to analyze how brain activation and wave frequencies occur in aphasic participants with preserved comprehension. Preliminary results indicate that the portable EEG device was able to adequately capture brain signals with the perspective of rehabilitation monitoring, which may support health professionals, especially speech-language pathologists, in rehabilitation centers.

In our sample, eight in nine of the participants showed increased electrical activation in the non-language dominant hemisphere. This may be revealing a migration of language, counter-lateral processing. This is a very interesting result because the idea is that “those who migrate the function to the other side recover more language”. Another interesting result was that the male participants showed less electrical activation in the right hemisphere, which may be revealing less language migration and perhaps less recovery.

Based on the experimental results, the proposed method has some positive implications, such as the possibility of graphical visualization of which areas are more active in the brain, and especially if there is activation in non-dominant areas of language, which may indicate improvement in the rehabilitation of the participant, including provoking greater involvement of the aphasic.

Another implication is the customization in the therapeutic conduct, that is, the speech therapist may customize the battery of tests in the therapeutic sessions, for each participant, based on previous sessions, the evolution and monitoring of brain activation.

We also believe that with the expansion of our sample, further analysis of the results, and inclusion of other linguistic tasks, new studies will emerge to improve the rehabilitation of aphasics, in the rehabilitation environment or even at home.

For this study, we were interested in analyzing brain activation of aphasic participants during a sentence completion task. Our next steps will include the adoption of three more tasks (word generation, rhyming, and object naming), to try to associate the EEG patterns with the pathology of aphasia.

The results obtained may be the starting point to use in the rehabilitation of aphasic individuals. In future studies we will extend the analysis to the other lobes of the brain to obtain a complete picture of oscillations. We hope to improve our experiment by creating a method that can try to predict the improvement of aphasics.

Supplemental Information

Supplemental Information 1 Raw data from 2021.

Electrophysiological signals collected from aphasic persons performing sentence completion tasks. Each file refers to an aphasic patient and contains the EEG (electroencephalogram) collection.

Click here for additional data file.

Supplemental Information 2 Raw data from 2022.

Electrophysiological signals collected from aphasic persons performing sentence completion tasks. Each file refers to an aphasic patient and contains the EEG (electroencephalogram) collection.

Click here for additional data file.

Additional Information and Declarations

Competing Interests

Author Contributions

Human Ethics

Data Availability

The authors declare that they have no competing interests.

Claudia Lima conceived and designed the experiments, performed the experiments, analyzed the data, prepared figures and/or tables, authored or reviewed drafts of the article, and approved the final draft.

Jeferson Andris Lopes analyzed the data, prepared figures and/or tables, and approved the final draft.

Victor Souza analyzed the data, authored or reviewed drafts of the article, and approved the final draft.

Sarah Barros conceived and designed the experiments, performed the experiments, analyzed the data, authored or reviewed drafts of the article, and approved the final draft.

Ingrid Winkler analyzed the data, authored or reviewed drafts of the article, and approved the final draft.

Valter Senna analyzed the data, authored or reviewed drafts of the article, and approved the final draft.

The following information was supplied relating to ethical approvals (i.e., approving body and any reference numbers):

The present research was approved by the Ethics Committee on Human Research of the Integrated Manufacturing and Technology Campus (CIMATEC)—Senai/Bahia (CAAE: 29622120.2.0000.9287) and approved by the Health Secretariat of the State of Bahia—SESAB (CAAE: 29622120.2.3001.0052) with the Center for Prevention and Rehabilitation of People with Disabilities-CEPRED, as a coparticipant center.

The following information was supplied regarding data availability:

The Aphasia EEG raw data is available in the Supplemental Files.

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
