# Peer review of "Analysis of brain activation and wave frequencies during a sentence completion task: a paradigm used with EEG in aphasic participants"

_PeerJ, doi:10.7717/peerj.15518_

## Round 0.1 · original submission · Major Revisions

Please, note carefully whether each reviewer's observations make sense for the authors. After, send us a new article version for a new valuation.
Regards.

·

Basic reporting

A copy desk is needed to check the punctuation, which is not satisfactory, and the need for extensive English revision.

You used two different styles of apostrophes or quotation marks in your document. Both types are acceptable, but it's best to be consistent.

The table needs to be verified.

The hypothesis as well as the problematization are not clear and concise. I suggest redoing it.

Experimental design

Methods should be described in more detail.
The sample should be more detailed.
The sample must be identified as participants and not as patients.
How the sample was selected.
What are the inclusion and exclusion criteria for the sample?
Could the female predominance be a bias for the results?

If there is a hypothesis, please describe the statistic used to answer your thesis. As well as using descriptive statistics to characterize the sample and justify the percentages used throughout the text.

Validity of the findings

The statistic needs details as to its procedure.
Reference the validation of the instrument used in the methods.

Additional comments

Dear Authors,
The study is very interesting and relevant to the area of knowledge.
Best regards,

Reviewer 2 ·

Basic reporting

See my additional comments.

Experimental design

See my additional comments.

Validity of the findings

See my additional comments.

Additional comments

The authors presented a study to analyze brain activation and wave frequencies of aphasic individuals during a sentence completion task, to possibly assist health professionals with the analysis of the aphasic subject's rehabilitation and task redefinition. Though the proposed analysis is interesting, some critical comments need to be addressed before acceptance for publication.
1. In a separate paragraph it is suggested to provide some remarks to further discuss the proposed methods, for example, what are the main advantages and limitations in comparison with existing methods?
2. I believe that it will make this paper stronger if the authors present some insightful implications based on their experimental outcomes. It would be helpful if the authors could interpret the association of EEG patterns with the pathology of aphasia.
3. Some closely related studies have recently been reported about advanced EEG analysis. The authors need to give more review of those studies, such as: Adaptive multi-model knowledge transfer matrix machine for EEG classification; Improving EEG decoding via clustering-based multi-task feature learning; Modern views of machine learning for precision psychiatry.
4. The authors may briefly discuss the potential limitations of the proposed method and what are the future research directions of this study. How other researchers can work on your study to continue this line of research?

---

## Round 0.2 · Minor Revisions

Dear Dr. Lima,

Thank you for your submission to PeerJ.

It is my opinion as the Academic Editor for your article - Analysis of brain activation and wave frequencies during a sentence completion task: a paradigm used with EEG in aphasic participants - that it requires a number of Minor Revisions.

My suggested changes and reviewer comments are shown below and on your article 'Overview' screen.
Please address these changes and resubmit. Although not a hard deadline please try to submit your revision within the next 21 days.

Yours sincerely.

·

Basic reporting

all recommendations for correcting the paper have been made. Be attentive to the minor's corrections in English grammar.

Experimental design

OK

Validity of the findings

OK

Additional comments

Proofread the period and quotation marks. When the quotation does not start but ends the sentence, the period is after the quotation marks.

Reviewer 2 ·

Basic reporting

See additional comments.

Experimental design

See additional comments.

Validity of the findings

See additional comments.

Additional comments

The authors have addressed my concerns accordingly, so I recommend accepting the manuscript in the current version.

---

## Round 0.3 · accepted · Accept

Dear Dr. Lima,

The original Academic Editor is no longer available and so I am making a decision in my capacity as Section Editor.

You have addressed the concerns of the reviewers, therefore I am happy to recommend your paper for publication.